# A 135-190 GHz Broadband Self-Biased Frequency Doubler using Four-Anode Schottky Diodes

**DOI:** 10.3390/mi10040277

**Published:** 2019-04-25

**Authors:** Chengkai Wu, Yong Zhang, Jianhang Cui, Yukun Li, Yuehang Xu, Ruimin Xu

**Affiliations:** School of Electronic Science and Engineering, University of Electronic Science and Technology of China, Chengdu 611731, China; chengkaiwu@std.uestc.edu.cn (C.W.); cuijianhang@std.uestc.edu.cn (J.C.); 18215605919@163.com (Y.L.); yuehangxu@uestc.edu.cn (Y.X.); rmxu@uestc.edu.cn (R.X.)

**Keywords:** frequency doubler, broadband matching, Schottky diodes, self-bias resistor, conversion loss, three-dimensional electromagnetic (3D-EM) model, millimeter wave, terahertz

## Abstract

This paper describes the design and demonstration of a 135–190 GHz self-biased broadband frequency doubler based on planar Schottky diodes. Unlike traditional bias schemes, the diodes are biased in resistive mode by a self-bias resistor; thus, no additional bias voltage is needed for the doubler. The Schottky diodes in this verification are micron-scaled devices with an anode area of 6.6 μm^2^ and an epitaxial layer thickness of 0.26 μm. For accurate design of the doubler, the 3D-EM model of the Schottky diode is built up to extract the parasitic parameters induced by the diode package when frequency rises up to the terahertz band. In order to implement broadband working, input waveguide steps, output suspended microstrip steps, and output probe with bias filter are all used as matching elements for impedance matching. Measured results show that the doubler exhibits a 3 dB bandwidth of 34% from 135 GHz to 190 GHz, with a conversion efficiency of above 4% when supplied with 100 mW of input power. A 17.8 mW peak output power with a 10.2% efficiency was measured at 166 GHz when the input power was 174 mW. The measured results agree well with the simulated results, which indicates that the self-bias scheme for Schottky diode-based frequency multipliers is feasible and effective.

## 1. Introduction

As the last spectrum resource that has not been fully exploited and utilized, the terahertz wave has many potential applications, such as high-speed communication, biomedicine, radio astronomy, and safety imaging [1,2]. The tremendous interest in the terahertz spectrum has prompted researchers to develop a series of components working in the terahertz band, such as amplifiers [3,4], frequency multipliers [5], mixers [6], antennas [7,8,9,10] and so on, to construct practical terahertz front-end systems [11,12,13]. As a key component of terahertz heterodyne systems, frequency multipliers with high enough power are required as front-stage drivers of terahertz frequency multiplier chains [14] or local oscillators (LO) of terahertz mixers. Theoretically, any non-linear device can be used to achieve the function of frequency multiplication by extracting the required harmonic components. Frequency multiplication technology based on planar Schottky diodes is one of the most attractive device technologies to produce millimeter wave/terahertz waves due to its high reliability, high spectral quality, low cost, and operation at room temperature [15,16]. Moreover, great progress has been made in the fabrication and modeling technology of Schottky diodes [14,17,18] as well as in the understanding and characterization of inner physical mechanisms [19,20,21].

Frequency multipliers mainly focus on the performance of efficiency, output power and bandwidth, and various state-of-the-art multipliers [22,23,24,25,26,27,28,29,30,31,32] based on planar Schottky barrier diodes that have been designed and demonstrated in recent years. Though frequency multipliers like in [22,23] eliminate possible assembly errors and feature better consistency with microwave monolithic integrated circuit (MMIC) technology, the discrete diodes [24,25,26,27,28,29,30,31,32] are also applied for its low cost and reliable alignment process when the frequency is below 400 GHz [27]. From the perspective of bias approaches, all these frequency multipliers mostly work at zero bias or external reverse bias. As we know, the nonlinear effect of diodes generates not only harmonic components, but also a DC component. However, there are few reports in the literature on a diode’s own DC component being used to bias itself in the terahertz band. Moreover, previous experience tells us that frequency multipliers can also work in a self-bias state, which inspired us to design a frequency multiplier that works directly under self-bias conditions.

To demonstrate the self-bias scheme, a 135–190 GHz frequency doubler based on Schottky diodes was designed, in which a resistor parallel to the ground in the bias circuit is utilized to withstand the DC component generated by the diodes to form a self-bias loop. An accurate three-dimensional electromagnetic (3-D EM) model of the diodes with a self-bias resistor was established using the finite element method (FEM), and the influence of the self-bias resistor on the performance of the frequency doubler was also investigated through a harmonic balance simulation. In order to achieve broadband impedance-matching, the optimum embedded impedance of the diodes was extracted using harmonic load-pull. Based on that, the input and output matching circuits were designed. Finally, the doubler, which worked under self-bias condition and exhibited the merits of broad band, high efficiency and high output power, was fabricated, assembled and measured.

## 2. Materials and Methods

### 2.1. Diode Modeling

Frequency multipliers based on Schottky diodes use the nonlinear effect of the Schottky barrier junction to produce the required harmonics. In our design, the varactor diode chip, which was produced by Teratech (Oxford, UK) had four anodes arranged in anti-series, as depicted in Figure 1a, and the overall chip dimensions were 420 μm × 80 μm × 50 μm. The diodes used an epitaxial layer with a doping density of 2 × 10^17^ cm^−3^, a diode anode size of 6.6 μm^2^ and an epitaxial thickness of 0.26 μm, resulting in a calculated zero-bias capacitance of 9.8 fF, a series resistance of 4.1 Ω, and a corresponding cut-off frequency 2.2 THz. These diodes were fabricated on GaAs substrate and had a gold air-bridged structure designed to minimize the parasitic capacitance.

Figure 1c shows the 3-dimentional (3-D) cross-sectional view of the diodes, including the air-bridge anode finger, the SiO_2_ passivation layer, the light doped n-GaAs epitaxy layer, the heavily doped n++ GaAs buffer layer, and the undoped GaAs substrate from top to bottom. Some approximation processing was needed because the field simulation software could not simulate the doping of the semiconductor. Some detailed information on the thickness and material of various layers are given in Table 1. According to the electromagnetic characteristics of the materials, a 3-D structure model of the diodes was established with appropriate material and size settings, as shown in Figure 1b. In addition, some key SPICE parameters of the diodes are given in Table 2 for the characterization of the intrinsic part of the Schottky junction.

When frequency rose up to the submillimeter wave band, the complicated parasitic effects induced by diode package could not be ignored and the cavity effect of complex structures also had to be taken into consideration. The accurate three-dimensional electromagnetic model (3-D EM) of the diodes in a balanced structure was built up in a High Frequency Structure Simulator (HFSS) (ANSYS, Canonsburg, PA, USA, 2015) [33] to extract the parasitic parameters and simulate the electromagnetic environment where the diode chip was located, as shown in Figure 2. Since HFSS does not support enclosed wave ports, coaxial probe technique [17] was adopted in the process of modelling and the anode probe penetrated through the epitaxial layer to the buffer layer to set internal wave ports at the interface, as depicted in Figure 1d.

The equivalent circuit model of the diodes was constructed with the parasitic parameters derived from the three-dimensional electromagnetic model of SNP items and the intrinsic SPICE parameters of the Schottky barrier, as shown in Figure 3. In the equivalent circuit model, a resistor was introduced to form a self-bias loop. The method of harmonic load-pull was then executed using Agilent’s Advanced Design System (ADS) (Agilent, Palo Alto, CA, USA, 2013) [34] on the equivalent circuit model to determine the optimum embedding impedance, which was 227.3-j24.2 Ω for the input at 80 GHz and 34.6-j26.8 Ω for the output at 160 GHz, respectively. In carrying out the load-pull procedure, the self-bias resistor was replaced by a DC voltage source for convenience. The optimum bias voltage was found to be −2 V for 100 mW input power, with a conversion efficiency of 20%.

### 2.2. Doubler Design

In a balanced structure, frequency multipliers have only an even-order or odd-order harmonics output; thus, there is no need for additional filters which tend to add circuit loss and degrade the multiplication bandwidth. In the balanced frequency doubler proposed originally by Erikson [35,36], the Schottky diode in anti-series configuration was placed at the junction where the balanced transmission line converted to unbalanced. In this structure, the incident fundamental wave was coupled onto the anti-series Schottky diode pair in a TE10 mode of the input waveguide and prevented from propagating into the output circuit by the backshort, which was achieved via narrowing down the input waveguide width. The even-order harmonics were then free to propagate down the output circuit in a TEM mode, while the odd-order harmonics were suppressed. The isolation between the input and output circuit in the balanced doubler was naturally realized by the electromagnetic mode orthogonality.d

According to these parameters, a 135–190 GHz broadband self-biased frequency doubler was proposed, as shown in Figure 4. The doubler was composed of input waveguide steps, output suspended microstrip steps, an output probe, bias filter, Schottky diode model, and a self-bias resistor. The input waveguide was WR10 and the output waveguide was WR5.1. The fundamental wave energy coupled to the diodes generated not only an even harmonic output, but also a DC component. The DC was loaded on the self-bias resistor, which in turn provided bias for the diodes via the bias filter. For the convenience of subsequent measurements, this resistor was placed outside the cavity. In practical applications, it could be easily integrated into the cavity. The external self-bias resistor was connected to the internal bias network through a piercing capacitor fixed on the cavity and a bias line made of Rogers 5880 substrate with a thickness of 0.127 mm. In order to prevent second harmonic leakage, a two-order Common Mode Resonant Cell (CMRC) structure was used in the bias circuit, which formed two transmission zeroes in the output frequency band, and thus achieved good isolation between the output frequency and DC bias. The output circuit adopted a suspended microstrip rather than a microstrip for two reasons. On the one hand, the suspended microstrip can reduce the loss of the matching circuit, considering that most of its fields are distributed in the air. On the other hand, compared with the microstrip, the suspended microstrip can improve the cut-off frequency of the higher-modes and guarantee that the second harmonic propagates in a single mode. The output suspended microstrips, output probe, and bias filter were integrated on a 50 μm thick quartz substrate. In addition, two steps with a depth of 70 μm were added on both sides of the reduced height input waveguide underneath the diodes in order to support the quartz substrate, while the other side of quartz substrate was supported by the ground of the CMRC microstrip.

In order to investigate the impedance characteristics of the diodes in a wide frequency band, load-pull analysis was performed on the equivalent circuit model, as shown in Figure 3, to give12% efficiency at different input frequencies under the optimum bias voltage. The results of load-pull at different frequencies were sketched on the same Smith chart shown in Figure 5, which shows that equal efficiency circles rotate counter-clockwise on the Smith chart as the frequency increases from 75 to 90 GHz and an intersection region is formed. The intersection of equal efficiency circles at different input frequencies is the broadband output matching region, where the efficiency exceeds 12% for the 75–90 GHz input frequency range, provided that the output load impedance falls into the region. Analogously, the broadband input matching region could also be obtained by a similar source-pull operation. The existence of the broadband matching region indicated that the varactor was biased in resistive mode and that broadband input matching and output matching were theoretically achievable.

Figure 6 shows the design of input and output matching. The input matching circuit included two-stage reduced height waveguides, while the height of the second stage waveguide was fully reduced to suppress the unwanted TM11 mode. The output matching circuit included not only suspended microstrip steps, but also the output waveguide E-plane probe, which took part in output matching while realizing the mode conversion (from TEM to TE10). The S-parameters of the waveguide steps, suspended microstrip steps and E-plane waveguide probe were analyzed and extracted in SNP files using HFSS to characterize the discontinuity effects. Moreover, the suspended microstrip model was also constructed by extracting feature parameters in the field simulation, such as characteristic impedance, attenuation constant and effective permittivity. Based on the above work, the whole harmonic balance simulation circuit of the frequency doubler was established. L1–L7 in Figure 6 were the main matching variables in the whole circuit, which were adjusted and optimized iteratively by HFSS in combination with ADS, so that Zin on the reference plane 1-1’ and Zout on the reference plane 2-2’ fell into the broadband matching region in the required bandwidth. Thus, broadband input and output matching was implemented.

The DC voltage source was then replaced with the self-bias resistor. Simulation results showed that conversion efficiency ascended with the increase of the self-bias resistor from 10 to 100 Ω, while it declined with the increase of self-bias resistance from 100 to 400 Ω, as shown in Figure 7a,b, respectively. The efficiency curve agreed well with the simulation results under the optimum bias voltage, especially when the self-bias resistor was 100 Ω. Therefore, it could be speculated that the optimum self-bias resistor was 100 Ω. Figure 8 shows the simulated results of input return loss, conversion loss and output power with 100 mW input power under a 100 Ω self-bias resistor.

### 2.3. Fabrication and Measurement

The manufacturing of the whole frequency doubler required numerically controlled metal machining (CNC) to process the cavity structure, thin film circuit technology to process the quartz substrates, and printed circuit board (PCB) technology to make the 5880 bias lines. After finishing the design of the doubler, the fabricating maps of the planar circuit and cavity structure were drawn by AutoCAD (2016) and Solidworks (2016). The whole frequency doubler circuit, including the bias filter, was manufactured on a 50 μm thick quartz substrate, with dimensions of 2.8 mm × 0.52 mm, embedded in the split waveguide block made of brass plated with 2 μm thickness of gold layer. The assembly procedures of the doubler mainly included cavity cleaning, substrate adhering, diode adhering, circuit sintering, capacitor installation, and wire bonding. In the assembly, substrate alignment was very important, otherwise the performance would be greatly deteriorated by substrate offset. Besides, it should be noted that the baking temperature did not exceed 200 °C to prevent the substrate from bending due to excessive temperature. The internal structure of the assembled frequency doubler is shown in Figure 9. The diode chip was mounted onto the quartz substrate circuit with the cathode pads connected to the split waveguide block, using the conductive silver adhesives to realize the DC and RF grounding, and to provide a good heat dissipation channel. The bias filter on the quartz substrate was connected to a bias line made of Rogers 5880 substrate, using a wire bond to provide a bias path.

For the purpose of verifying the proposed the self-bias scheme for Schottky diodes, two prototypes of the frequency doubler were manufactured for comparative measurements. The test platform of the doublers is shown in Figure 10. In order to measure the frequency doublers by turn, the test frequency band was divided into three parts: 130–160 GHz, 160–180 GHz, and 180–200 GHz bands. Driving power was provided by an Agilent E8257D PSG Analog Signal Generator (Santa Clara, CA, USA), followed by different MMIC frequency multiplication and amplification modules. The input power was calibrated in advance by the VDI Erickson PM4 millimeter/submillimeter power meter (Virginia Diodes, Inc., Charlottesville, VA, USA). The output power of the doublers was measured using the same power meter, together with a 2.5 cm long WR5.1 to WR10 waveguide transition to minimize the mismatch between the doubler and meter (the measured output power was not corrected for the transition loss). An external potentiometer was used as a self-bias resistor, which could be easily adjusted to find the optimum self-bias resistance.

## 3. Results and Discussion

The measured conversion efficiency and output power are shown in Figure 11. A conversion efficiency of around 8% and a mean output power of 8 mW from 140 to 180 GHz were measured under a 100 Ω self-bias resistor with 100 mW input power. The doubler exhibited a 3 dB conversion loss bandwidth of 34% from 135 GHz to 190 GHz. The self-bias voltage generated by the diodes under 100 Ω self-bias resistor was measured close to −2 V when the input power was 100 mW, which was in accordance with the simulation results. As the input power increased, the self-bias voltage decreased slightly.

When the maximum input power was provided, the frequency doubler was tested under different self-bias conditions, as shown in Figure 12. The output power increased with the increase of the self-resistor from 10 to 100 Ω and it decreased distinctly with the increase of the self-bias resistor from 100 to 400 Ω, with 160 mW input power at 160 GHz. A similar phenomena occurred with a peak output power of 17.8 mW and a peak efficiency of 10.2% at 166 GHz when the input power was 174 mW, except that the optimum self-bias resistor was 200 Ω. This was a slight deviation from the simulation results shown in Figure 8, and was probably caused by a power saturation effect, because each anode shared more than 40 mW input power. The measured output power and efficiency as a function of input power under the 50 Ω self-bias resistor are shown in Figure 13. The output power increased with the input power and the efficiency was stable at 8%. If greater power was provided, the doubler would have a higher output power level at both ends of the working frequency band. It should be noted that the curves in Figure 12 and Figure 13 represent the average values of the measured results of the two prototypes, and the error bars represent the standard deviations of the measured results for each prototype.

A comprehensive comparison of the designed frequency doubler with the multipliers reported in the literature is given in Table 3. Compared with the tripler [24] and the doublers [25,26,27] working under external bias, the proposed doubler had a larger fractional bandwidth. Although a record full waveguide bandwidth of 44% was obtained by the doubler working under zero bias in ref. [29], its output power and power capacity were smaller than this work due to the fewer number of anodes. The doubler working under zero bias in Ref. [22] used the largest number of anodes in the whole table and achieved a similar efficiency as shown in this work. However, higher output power and larger fractional bandwidth were obtained by the proposed doubler using only four anodes. From the above comparison, it can be concluded that the proposed self-bias scheme for frequency multipliers achieved a compromise in bandwidth and efficiency. For the designed doubler itself, the combination of broadband and high output power made it versatile.

## 4. Conclusions

In this article, a 135–190 GHz broadband self-biased frequency doubler based on planar Schottky diodes was demonstrated. A resistor was utilized to withstand the DC component generated by the diodes to form a self-bias scheme, which contributed to simplifying the bias scheme of the frequency multipliers. Meanwhile, input waveguide steps, output suspended microstrip steps and an output probe with bias filter were all used as matching elements for broadband impedance matching. The validity and accuracy of the diode model were verified by the agreement between the simulation and measurements. The frequency doubler presented in this work has the merits of broad band, high efficiency, high output power and self-bias working, which makes it very attractive for practical broadband terahertz applications.

## Figures and Tables

**Figure 1 micromachines-10-00277-f001:**
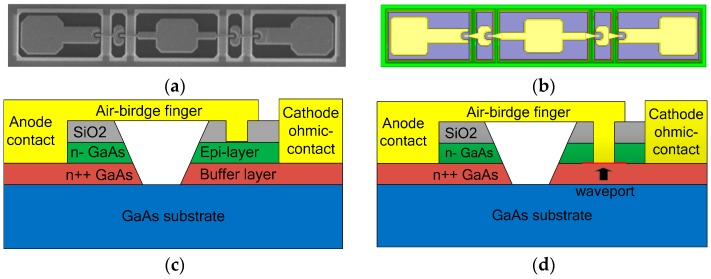
Diode structure and internal port (**a**) SEM Image of the Teratech GaAs anti-series air-bridged Schottky diodes; (**b**) 3-D structure model of the diodes; (**c**) 3-D cross-sectional view of the diodes; (**d**) internal port setting.

**Figure 2 micromachines-10-00277-f002:**
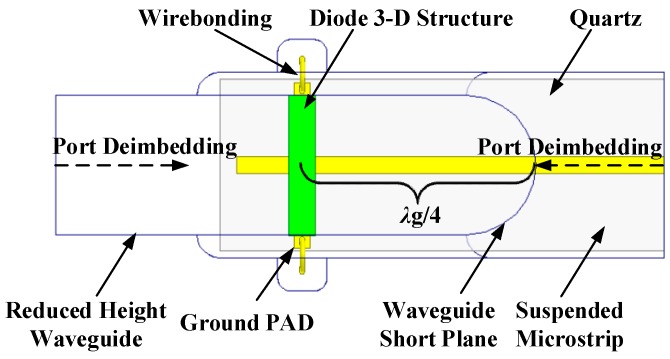
The 3-D EM model of the diodes in balanced structure in HFSS.

**Figure 3 micromachines-10-00277-f003:**
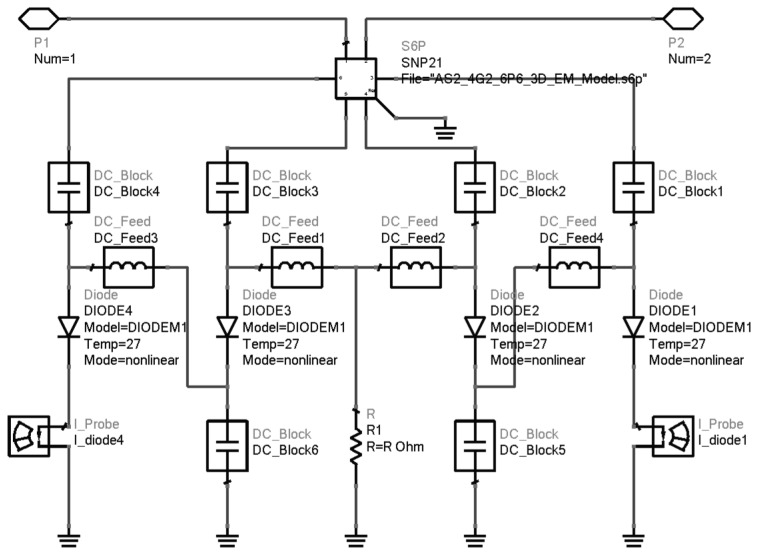
The equivalent circuit model of the diodes with self-bias resistor in ADS.

**Figure 4 micromachines-10-00277-f004:**
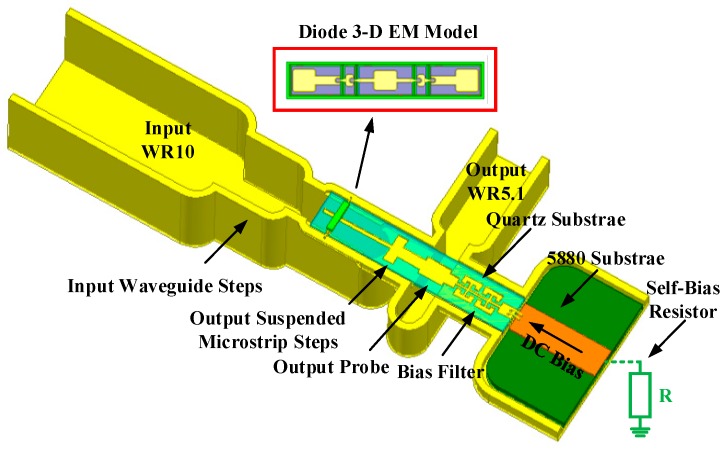
The configuration of the 135-190 GHz balanced frequency doubler under self-bias.

**Figure 5 micromachines-10-00277-f005:**
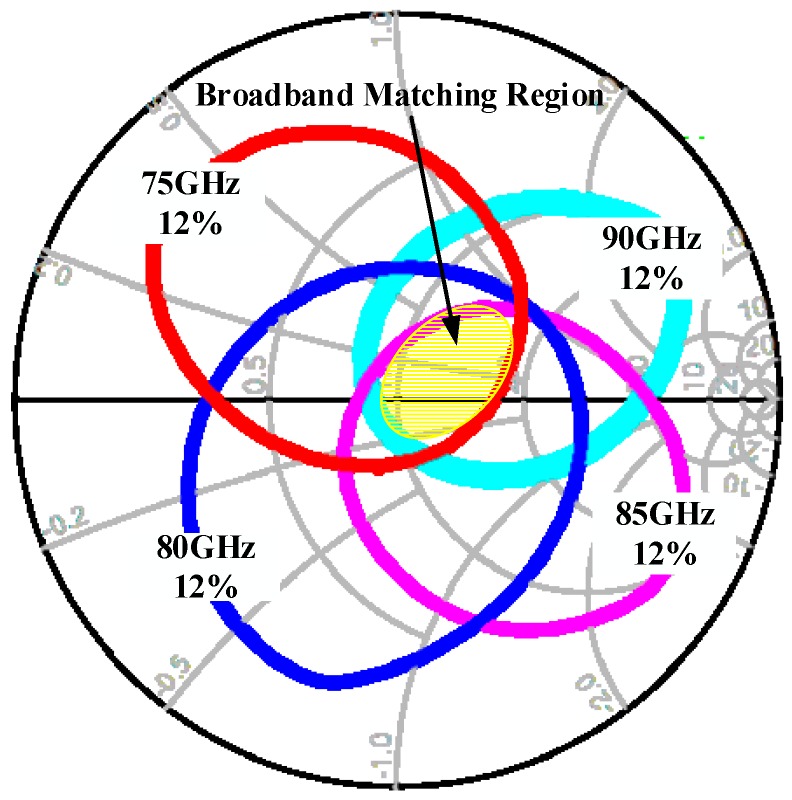
Output load-pull for a 12% efficiency at different input frequencies.

**Figure 6 micromachines-10-00277-f006:**
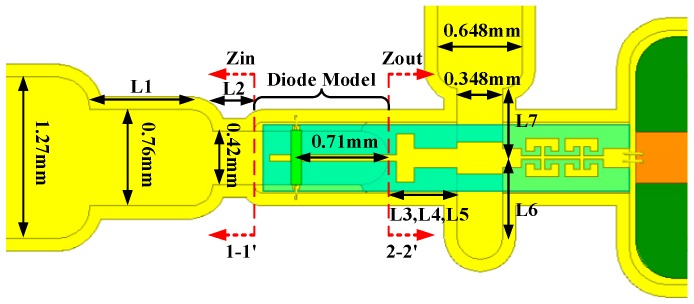
Design of input and output matching (L1 = 767 μm, L2 = 318 μm, L3 = 58 μm, L4 = 138 μm, L5 = 330 μm, L6 = 622 μm, L7 = 547 μm).

**Figure 7 micromachines-10-00277-f007:**
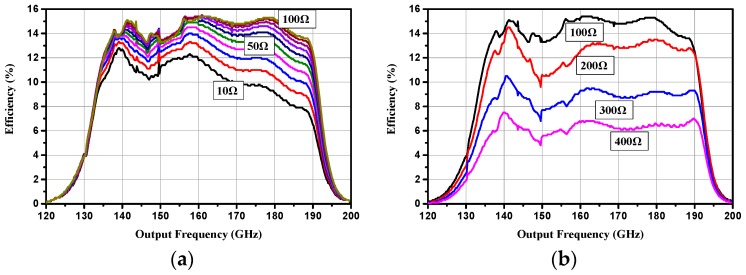
Simulated efficiency under different self-bias resistors (**a**) from 10 to 100 Ω; (**b**) from 100 to 400 Ω.

**Figure 8 micromachines-10-00277-f008:**
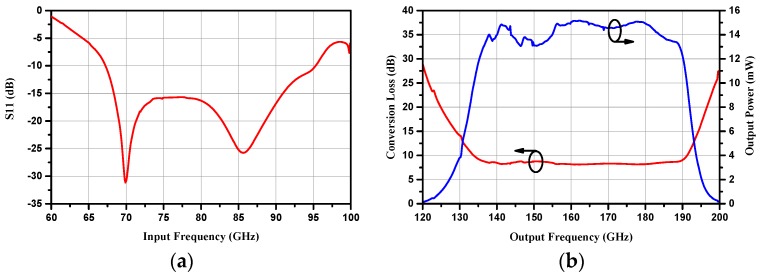
Simulated input return loss, conversion loss and output power with 100 mW input power under a 100 Ω self-bias resistor.

**Figure 9 micromachines-10-00277-f009:**
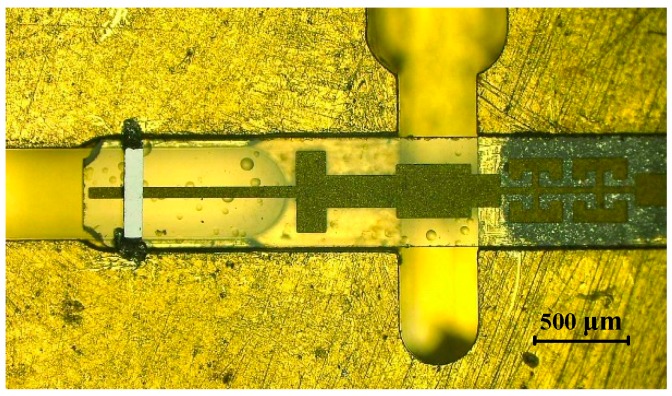
The image of the 135–190 GHz doubler imbedded in the split-waveguide block.

**Figure 10 micromachines-10-00277-f010:**
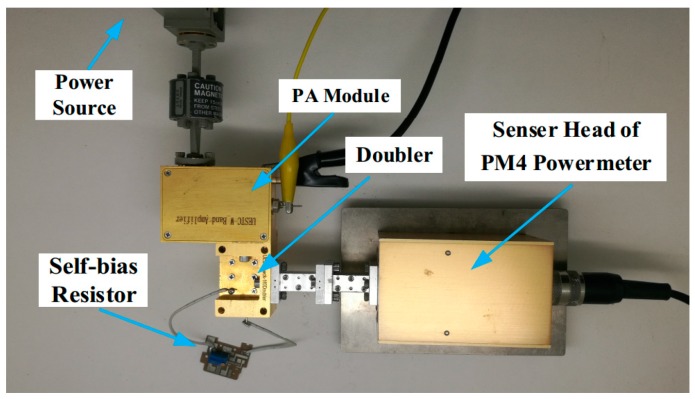
Test platform of the doubler.

**Figure 11 micromachines-10-00277-f011:**
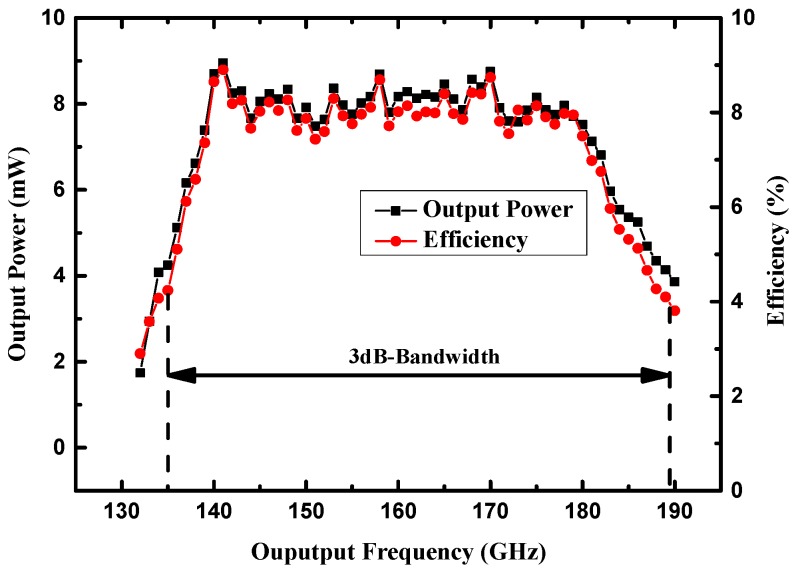
Measured efficiency and output power of the doubler under a self-bias resistor of 100 Ω.

**Figure 12 micromachines-10-00277-f012:**
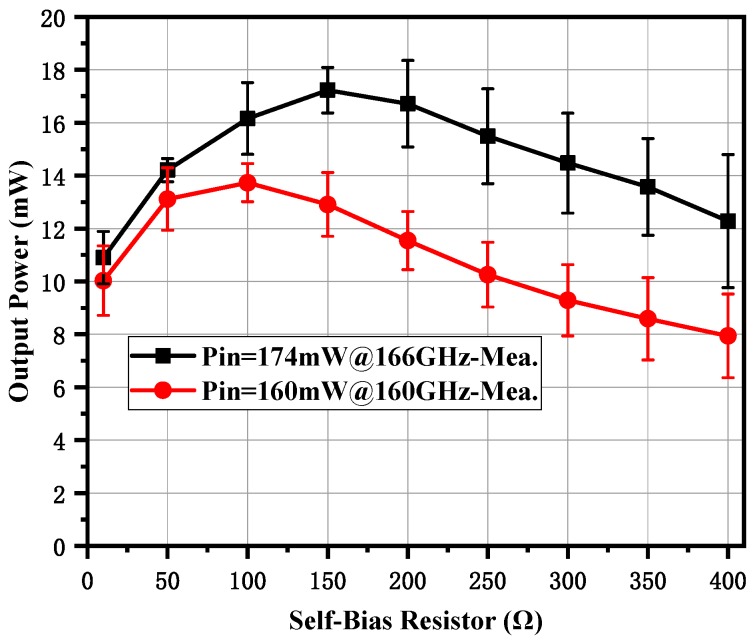
Measured output power of the doubler under different self-bias resistors with the maximum input power.

**Figure 13 micromachines-10-00277-f013:**
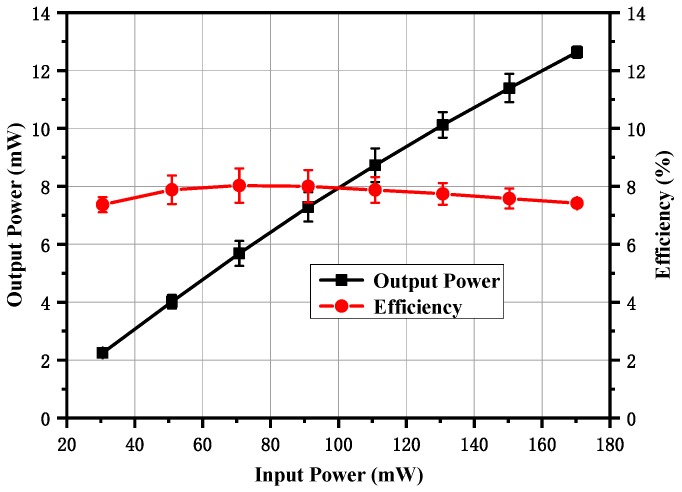
Measured output power and efficiency versus input power at 166 GHz under 50 Ω self-bias resistor.

**Table 1 micromachines-10-00277-t001:** Layer Structure Parameters of the diodes used in this paper.

Layers of the Diodes	Thickness (μm)	Doping Concentration (cm^−3^)	Material	Relative Dielectric Constant
Semi-Insulating Substrate	50	-	GaAs	12.9
Buffer Layer	5	5 × 10^18^	Pec	1
Epi-Layer	0.26	2 × 10^17^	GaAs	12.9
Oxide Layer	0.5	-	SiO_2_	4
Ohmic Contact Layer	0.76	-	Gold	1
Anode and Cathode	1	-	Gold	1

**Table 2 micromachines-10-00277-t002:** Some key SPICE parameters of the diodes.

Parameters	Value
Reverse Saturation Current, *I_s_*	9.39 × 10^−15^ A
Series Resistance, *R_s_*	4.1 Ω
Ideal Factor, *N*	1.12
Zero-biased Junction Capacitance, *C_j0_*	9.8 fF
Junction Potential, *V_j_*	0.85 V
Reverse Breakdown Current, *I_br_*	1 μA
Energy Gap, *E_g_*	1.43 eV

**Table 3 micromachines-10-00277-t003:** Comprehensive comparison of reported frequency multipliers.

Ref.	Frequency (GHz)	Number of Anodes	Pin (mW)	Pout (mW)	Effi. (%)	FBW (%)	Bias Way
[22]	177–202	6 × 2	20–120	1–12	4.3–10.2	22	Zero Bias
[24]	200–235	6	20–60	1.5–15	~16	24	External Bias
[25]	150–170	6	10–200	1–35	5.1–23.5	10	External Bias
[26]	190–198	6	100–260	3.7–20	1.4–8	4	External Bias
[27]	180–200	6	50–95	4–13	4–10	10.5	External Bias
[29]	140–220	2	20–32	1.8–3.7	8.7–12.7	44	Zero Bias
This Work	135–190	4	30–174	2.1–17.8	4–10.2	34	Self Bias

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
