# Peer review of "A 135-190 GHz Broadband Self-Biased Frequency Doubler using Four-Anode Schottky Diodes"

_micromachines, 2019, doi:10.3390/mi10040277_

Round 1
Reviewer 1 Report
This paper presents self-biased frequency doubler using four-anode Schottky diodes. A resistor was used to withstand the DC component generated by the diodes to form a self-bias scheme. Considering the importance of diodes in electrical engineering, this paper would be of interest to mechanical, electrical engineering, and electronics fields. However, several comments below must be addressed before reconsideration in 'Micromachines'
1) cross-sectional SEM or TEM images for Fig. 1(b) should be provided to demonstrate epi layers on GaAs substrate and demonstrate overall layers for structure.
2) Reliability issue: data in Fig. 12 and 13 must have error bars for each data.
3) Authors need to provide the fabrication methods in detail.
Author Response
The revisions of the manuscript entitled “A 135-190 GHz Broadband Self-Biased Frequency Doubler using Four-Anode Schottky Diodes”, corresponding manuscript ID: micromachines-479447, have been completed according to the reviewers’ comments. We sincerely thank the editor and all reviewers for their valuable feedback that we have used to improve the quality of our manuscript. The response to the reviewers is attached in the following word file.

Reviewer 2 Report
An interesting topic has presented by authors in their research work. The paper has well written and has well organized. The figures with high quality have presented. The authors have provided a comparison section to compare their work with the literature, which caused to increase the validity of the proposed article. I suggest authors to address step by step my following comments before my recommendation to publish their paper.
1) Please add one sentence about novelty of the proposed work in the abstract section. The dimensions of the proposed structure and its practical applications could be interesting if you can add them in this part as well.
2) Please add one paragraph in the introduction section regarding what you did in this work?
3) To create more comprehensives of your paper, you can briefly explain other microwave components such as antennas, amplifiers, and so on, which are working on terahertz band, at very least you can add some references. Following references are suggested.
“Advances in terahertz communications accelerated by photonics”, Nature Photonics, vol. 10, pp. 371-379, June 2016.
“Beam-Scanning Leaky-Wave Antenna Based on CRLH-Metamaterial for Millimeter-Wave Applications”, IET Microwaves, Antennas & Propagation, 6pp. DOI: 10.1049/iet-map.2018.5101, Available online: 06 March 2019.
“Design and Fabrication of a 1 THz Backward Wave Amplifier”, Terahertz Science and Technology, Vol.4, No.4, pp.149-163, December 2011.
"Mutual Coupling Suppression Between Two Closely Placed Microstrip Patches Using EM-Bandgap Metamaterial Fractal Loading", IEEE Access, vol. 7, Page(s): 23606 – 23614, March 5, 2019.
“Towards THz communications — status in research, standardization and regulation”, J. Infrared Milli. Terahz Waves 35, 53–62 (2014).
"Interaction Between Closely Packed Array Antenna Elements Using Metasurface for Applications Such as MIMO Systems and Synthetic Aperture Radars", Radio Science, Volume 53, Issue 11, November 2018, Pages 1368-1381.
“A survey of millimeter wave communications (mmWave) for 5G: opportunities and challenges”, Wirel. Netw. 21, 2657–2676 (2015).
“Study on Isolation Improvement Between Closely Packed Patch Antenna Arrays Based on Fractal Metamaterial Electromagnetic Bandgap Structures”, IET Microwaves, Antennas & Propagation, Volume 12, Issue 14, 28 November 2018, p. 2241 – 2247.
“Audio signal transmission over THz communication channel using semiconductor modulator”, Electron. Lett. 40, 124–125 (2004).
“Periodic Array of Complementary Artificial Magnetic Conductor Metamaterials-Based Multiband Antennas for Broadband Wireless Transceivers” IET Microwaves, Antennas & Propagation, Volume 10, Issue 15, 10 December 2016, p. 1682 – 1691.
“Ultrawide bandwidth single channel 0.4 THz wireless link combining broadband quasi-optic photomixer and coherent detection”, IEEE Trans. Terahz Sci. Technol. 4, 328–337 (2014).
"UWB Antenna Based on SCRLH-TLs for Portable Wireless Devices", Microwave and Optical Technology Letters, Volume 58, Issue 1, January 2016, Pages: 69–71.
“Performance comparison of MIMO techniques for optical wireless communications in indoor environments.” IEEE Trans. Commun. 61, 733–742 (2013).
“Bandwidth Extension of Planar Antennas Using Embedded Slits for Reliable Multiband RF Communications”, AEUE Elsevier- International Journal of Electronics and Communications, Volume 70, Issue 7, July 2016, Pages 910–919.
“Large-scale antenna systems with hybrid analog and digital beamforming for millimeter wave 5G.” IEEE Commun. Mag. 53, 186–194 (2015).
"New Compact Antenna Based on Simplified CRLH-TL for UWB Wireless Communication Systems", International Journal of RF and Microwave Computer-Aided Engineering, Volume 26, Issue 3, March 2016, pages: 217–225.
“120-GHz-band 20-Gbit/s transmitter and receiver MMICs using quadrature phase shift keying”, In Proc. 2012 7th European Microwave Integrated Circuit Conf. (EuMIC) 313–316 (IEEE, 2012).
“Metamaterial-Based Antennas for Integration in UWB Transceivers and Portable Microwave Handsets” International Journal of RF and Microwave Computer-Aided Engineering, Volume 26, Issue 1, January 2016, pages: 88–96.
“120-GHz-band wireless link technologies for outdoor 10-Gbit/s data transmission”, IEEE Trans. Microw. Theory Tech. 60, 881–895 (2012).
4) In section 2.2 (doubler design) please provide more explanations.
5) The authors have provided a nice comparison with the literature in the Table I. However, the presented explanations on this table is not enough. In other words, please provide more illustrations on this table by step by step comparing your paper with the mentioned references and provide a comprehensive paragraph to explain everything.
Author Response

(The authors gave the same response as above.)

Round 2
Reviewer 1 Report
Authors could not provide the detailed epi layers image even though knowing the exact value of each thickness is very important for the modeling. Also, the amplitude of error bars provided by authors in Fig. 12 and 13 looks identical for each data. How is the error bars' amplitude same? This reviewer think the data is unreliable. Thus, this review cannot recommend this manuscript for the publication.
Author Response
Thanks for your valuable comments on our manuscript. A top view of SEM image of the diodes is provided as well as some detail information on the thickness and material of various layers in the table 1. Table 2 summary some key SPICE parameters of the diodes.We're sorry that the treatment toward error bars was wrong, which is corrected based on the measured results of two prototypes of the doubler. The response to the reviewers is attached in the following word file.
